# Novel and effective plasmid transfection protocols for functional analysis of genetic elements in human cardiac fibroblasts

**Makoto Matsuyama**[1]*, **Takahiro Iwamiya**[1,2,3]

**1** Research & Development Department, Metcela Inc., Kanagawa, Japan, **2** Institute for Advanced Biosciences, Keio University, Yamagata, Japan, **3** Nagoya University Graduate School of Medicine, Nagoya, Japan

* m.matsuyama@metcela.com

## Abstract

Cardiac fibroblasts, have lower gene transfer efficiency compared to dermal fibroblasts, posing challenges for plasmid-based gene transfer methods. A higher transfer efficiency could enable improved insight into heart pathology and development of novel therapeutic targets. In this study we compared eleven commercially available transfection reagents and eight plasmid purification methods. Finally, we systematically evaluated 150 unique transfection conditions (incubation times, addition of innate immune inhibitors, reagent to plasmid ratios etc) to optimize the methodology. The aim was to develop an optimized plasmid transfection protocol specifically tailored for primary human cardiac fibroblasts with high efficiency and minimal toxicity. While the actual transfection efficiency, indicated by the expression of fluorescent proteins, was less than 5%, our optimized protocol was sufficient for achieving significant gene expression levels needed for experimental applications such as luciferase enhancer-promoter assays. Leveraging our newly developed methodology, we could perform comprehensive profiling of nine viral and native enhancer/promoters, revealing regulatory sequences governing classical fibroblast marker (VIM) and resident cardiac fibroblast marker (TCF21) expression. We believe that these findings can help advance many aspects of cardiovascular research. In conclusion, we here report for the first time a plasmid transfection protocol for cardiac fibroblasts with minimal cell toxicity and sufficient efficiency for functional genomic studies.

## Introduction

Fibroblasts have long served as invaluable resources for *in vitro* disease modeling and regenerative medicine research, in part due to their ease of primary culture. Mouse 3T3 cells were established in 1964 as non-cancerous immortalized cells and contributed to many aspects of viral research [1]. In the 1970s, researchers found that co-culturing fibroblasts with other cells increased the success rate of primary culture and started using fibroblasts as feeder cells [2,3].

And with the discovery of embryonic stem cells (ESCs) [4] and induced pluripotent stem cells (iPSC) [5], mouse embryonic fibroblasts (MEF) and human dermal fibroblasts (HDF)

**Data Availability Statement:** All relevant data are within the manuscript and its Supporting Information files.

**Funding:** The funder provided support in the form of salary for author [MM], but did not have any

additional role in the study design, data collection and analysis, decision to publish, or preparation of the manuscript. The specific roles of these authors are articulated in the 'author contributions' section.

**Competing interests:** I have read the journal's policy and the authors of this manuscript have the following competing interests: Takahiro Iwamiya is a cofounder and co-CEO of Metcela, Inc., and he has ownership of stocks. The corresponding author is an employee of Metcela. Takahiro Iwamiya has the authority to make payment decisions regarding employee salaries. Metcela is a company that develops VCAM1-expressing cardiac fibroblasts (VCFs) as a therapy for ischemic heart diseases. Additionally, Metcela applied for the following patents: Inventor: Takahiro Iwamiya. Assignee: Metcela Inc. Title: Composition For Injection Which Can Be Used For Treatment Of Heart Diseases And Contains Fibroblasts, And Method For Producing Fibroblast For Therapy Use. International application number: PCT/JP2018/006795. An object of the present invention is to present VCFs as a method that has not been established yet and that is useful for achieving long-term and fundamental cure of a necrotic cardiac tissue region to allow recovery of heart function. The funder provided support in the form of salary for author [MM], but did not have any additional role in the study design, data collection and analysis, decision to publish, or preparation of the manuscript. The specific roles of these authors are articulated in the 'author contributions' section. The authors declare a commercial affiliation with Metcela, Inc. This does not alter our adherence to PLOS ONE policies on sharing data and materials.

have been frequently used both as feeders and starting materials. Reprogramming fibroblasts into iPSCs essentially needs effective gene transfer technologies. While viral vectors initially dominated the introduction of the Yamanaka factors (Oct3/4, Sox2, Klf4, c-Myc) into HDF due to their high infection efficiency [5–11], their oncogenesis and insertional mutagenesis in infected cells is a bio-safety concern, and safer plasmid-based gene transfer methods in HDFs have been developed [12–16]. However, this transition has not been uniformly adopted in non-dermal fibroblasts, where reliance on viral vectors remains [17–19]. Therefore, the application of plasmid transfer into fibroblasts from diverse tissue origins is still largely unexplored.

For example, a technique has been proposed to induce cardiac fibroblasts to differentiate into cardiomyocytes by direct reprogramming [18], but a major hurdle is the extremely low gene transfer efficiency [20].

In light of the growing significance of establishing accessible plasmid transfection technology for non-skin fibroblasts, we specifically target human cardiac fibroblasts (HCFs). The aim of our study is to develop an optimized plasmid transfection protocol specifically tailored for primary human cardiac fibroblasts with enough efficiency and minimal toxicity. To achieve this, we conducted After a rigorous screening of commercially available transfection reagents and systematically evaluated 150 unique transfection conditions. W, we identified a protocol that achieves high transfection efficiency with minimal cytotoxicity.

To demonstrate the versatility of this technique, we applied it to luciferase enhancer-promoter assays and could successfully identify both exogenous and endogenous sequences that were robustly expressed in HCF.

The present study aims to address the critical need for a simple yet effective gene introduction method for cardiac fibroblasts, a cell type that has been challenging to manipulate genetically. Our objectives are threefold: (1) to establish an optimized plasmid transfection protocol for human cardiac fibroblasts (HCFs), (2) to elucidate the mechanisms underlying transfection-induced cytotoxicity, and (3) to demonstrate the utility of our method through functional analyses of genetic elements in HCFs. This approach has the potential to significantly advance our understanding of cardiac fibroblast biology. Furthermore, the identification of cardiac fibroblast-specific promoter and enhancer sequences could pave the way for targeted genetic manipulations, opening new avenues for both basic research and clinical applications in the field of cardiac biology.

## Materials and methods

All experimental protocols were approved by the Institutional animal care and use committee of Tokyo Women's Medical University and Metcela Inc. The permit numbers are GE21-064, GE22-073, and GE23-054.

### Cell culture

Human Cardiac-fibroblasts (HCFs) derived from the ventricle of an adult human heart (CAT #CC-2904, LOT #18TL281202) were purchased from Lonza (Basal, Switzerland) and cultured with HFDM-1(+) medium (Cell Science & Technology; Miyagi, Japan) supplemented with 1% (v/v) Newborn Calf Serum (NBCS/HyClone, Cytiva; Massachusetts, USA) and 1% penicillin and streptomycin. HCF cells at passage 8 or 9 were used in all experiments. Human Pulmonary-fibroblasts (CAT#C-12360, LOT#433Z024) were purchased from PromoCell (Heidelberg, Germany), and culture with NHDF-1/5%NBCS/1%PS. Human dermal-fibroblasts (CAT #C-12302, LOT#467Z026.3) were purchased from PromoCell, and cultured with NHDF-1/1% NBCS/1%PS. Human renal-fibroblasts (CAT#P10666, LOT#070518) were purchased from Innoprot (Bizkaia, Spain), and cultured with NHDF-1/5%NBCS/1%PS. Human cardiac-

fibroblasts derived from the auricle (CAT#KAC002-RAA01, LOT#YT-201208-01) were purchased from KAC (Kyoto, Japan). Lenti-X 293T cells were purchased from Takara-Bio (Shiga, Japan) to produce lentiviral vectors, cultured with DMEM high glucose (Fujifilm-Wako; Osaka, Japan) supplemented with 10% Fetal Bovine Serum (ThermoFisher Scientific; Massachusetts, USA) and 1% PS. All cells were cultured at 37°C in 5% $CO_2$ atmosphere.

## Plasmid construction

pUC19 purified by CsCl-EtBr ultracentrifugation was purchased from Takara-Bio. Short gene fragments were synthesized by gBlock service of Integrated DNA Technologies (Iowa, USA), and assembled into pLV-CMV-SNAPtag-DasherGFP, pGL4.14(Promoters, GLuc-IRES-DasherGFP-WPRE-SpA), and pGL4.14(Promoters, GLuc-WPRE-SpA) by InFusion Cloning (Takara-Bio).

## Mini-scale plasmid preparation

pUC19 purified by CsCl-EtBr was transformed into DH5a competent *E. coli* (DNA-903F, Toyobo; Osaka, Japan). Single colonies were picked up and inoculated in 3 mL LB Broth with Miller's modification (ThermoFisher Scientific) at 37°C shaking at 250 rpm. After 18-hour inoculation, plasmid was minipreped following the official protocols: FastGENE Plasmid Mini (Nihon Genetics; Tokyo, Japan), Monarch Plasmid miniprep kit (Promega; Wisconsin, USA), and NucleoSpin Plasmid Transfection-grade (Machrey-Nagel; Dueren, Germany). Miraprep protocol followed the publication [21], in which fresh RNase A (Nacalai Tesque; Kyoto, Japan) was added to the first resuspend solution on the day of miniprep. After the neutralization and centrifugation step, the same volume of ethanol was applied to clear lysates before binding to the columns. In Figs 5 and S2, all plasmids were minipreped by NucleoSpin Plasmid Transfection-grade kit.

## Midi-scale plasmid preparation

pLV-CMV-SNAPtag-DasherGFP was transformed into NEB Stable Competent E.coli (New England Biolabs; Massachusetts, USA) and a single colony was inoculated at 30°C in 200 mL Terrific Broth. It was midipreped with NucleoBond Xtra Midi EF (Machrey-Nagel). The final plasmid pellets were dissolved in TE buffer.

## Chemical transfection

When HCFs were transfected with pLV-CMV-SNAPtag-DasherGFP in Fig 1, 3.6e4 cells were seeded per well in a 24-well plate one day before transfection, and medium was changed just before the transfection to exclude dead cells. The protocols described below correspond to the volume of two wells in a 24-well plate, and in case of 96-well plates (Figs 2–5), the volume decreased in proportion to the surface area in a well.

**Lipofectamine3000 (Thermofischer Scientific) in Fig 1.** 1000 ng pLV-CMV-SNAPtag-DasherGFP plasmidand 2.0 μl P3000 reagent were added to 50 μl OptiMEM. 1.5 μl Lipofectamine3000 was added to 50 μl OptiMEM. Plasmid/P3000/OptiMEM was mixed thoroughly with Lipofetamine3000/OptiMEM, and incubated at room temperature for 10 minutes. 50 μl plasmid/P3000/Lipofectamine3000/OptiMEM mixture was equally applied to two wells in a 24-well plate. Twenty-four hours later, the medium was exchanged with fresh complete medium.

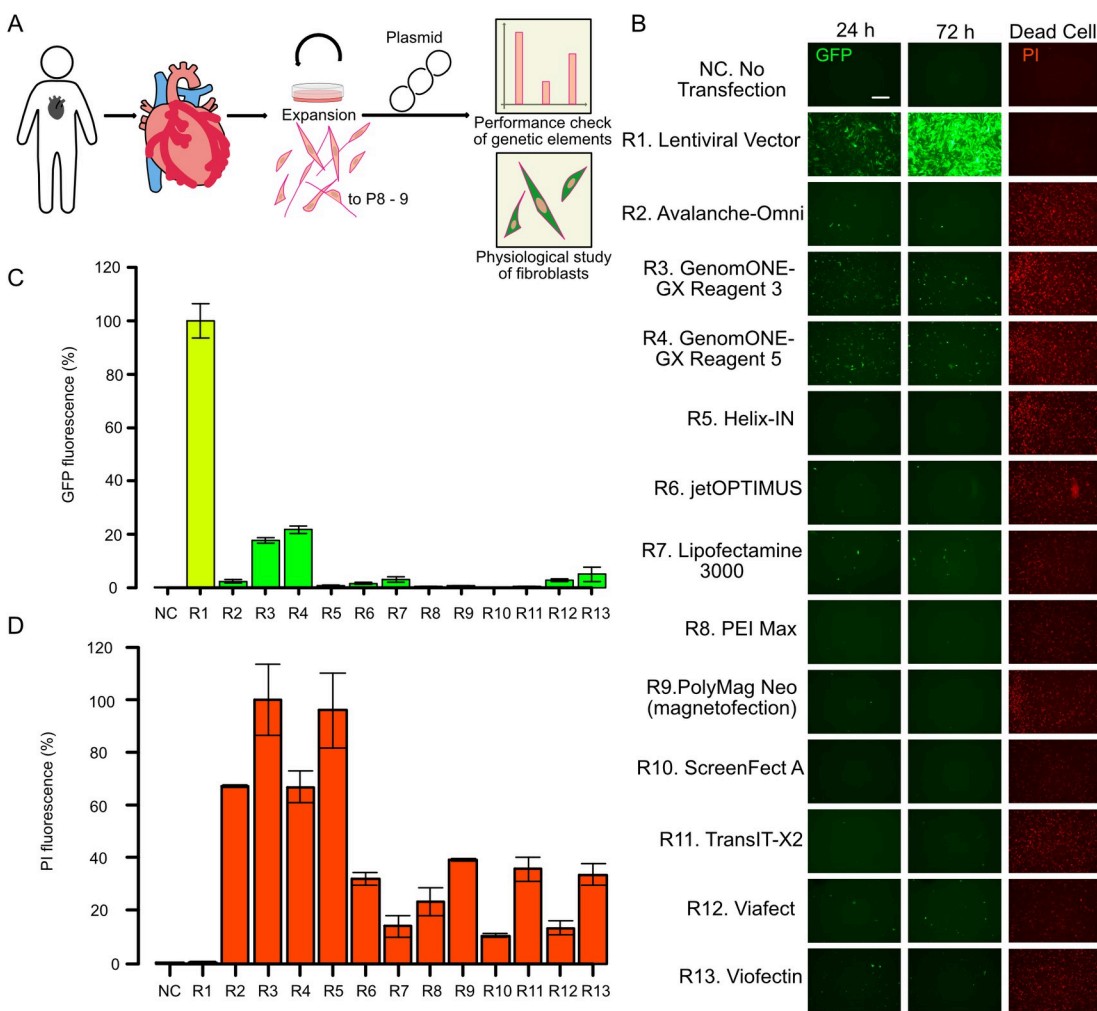

**Fig 1. Chemical transfection of primary human fibroblasts.** (A) Schematics of the assay procedure. Human fibroblasts were collected from human tissues and expanded up to passage 8 or 9. Various vectors were transfected for luciferase-based reporter assays and fluorescent protein-based protein localization studies. (B) A GFP-expressing plasmid was transfected into primary human cardio-fibroblasts using various reagents according to the manufacturers' protocols. A lentiviral vector was infected into fibroblasts as a positive reference. Cells were observed every 24 hours from the transduction until 72 h. After 72 h, propidium iodide (PI) was used to stain dead cells. Scale bar is 500 μm. (C) Fluorescent intensity comparison of GFP and PI in three biological replicates. GFP images taken 24 h after the transfection were used for analysis. PI images were taken at 72 h. The numbers 1–14 correspond to the reagent numbers in Fig 1B. The bar graph for the lentivirus data (positive control) is colored light green and light red in (C) and (D), respectively. Bar graphs represent the means of the biological triplicates, except for PI of R2, which is the mean of duplicate; error bars are the standard error of the mean.

All combinations of variable parameters (plasmid per well [100/150 ng/well], Lipofectamine3000 reagent per well [0.15/0.30/0.45 μl/well], and first medium change timing [4/8/24 hours after the transfection]) were tested in Fig 4.

**TransIT-X2 Dynamic Delivery System (Mirus Bio; Wisconsin, USA) in Fig 1.** TransIT-X2 reagent was stored at -20˚C, put in a safety cabinet to warm to room temperature and vortexed shortly just before use. 1000 ng pLV-CMV-SNAPtag-DasherGFP was added to OptiMEM 100 uL. 3 μl TransIT-X2 reagent was added to the Plasmid/OptiMEM, mixed well, and incubated at room temperature for 30 minutes, then 50 μl plasmid/TransIT-X2/OptiMEM mixture were equally applied to two wells in a 24-well plate. Twenty-four hours later, the medium was removed and exchanged with fresh complete medium.

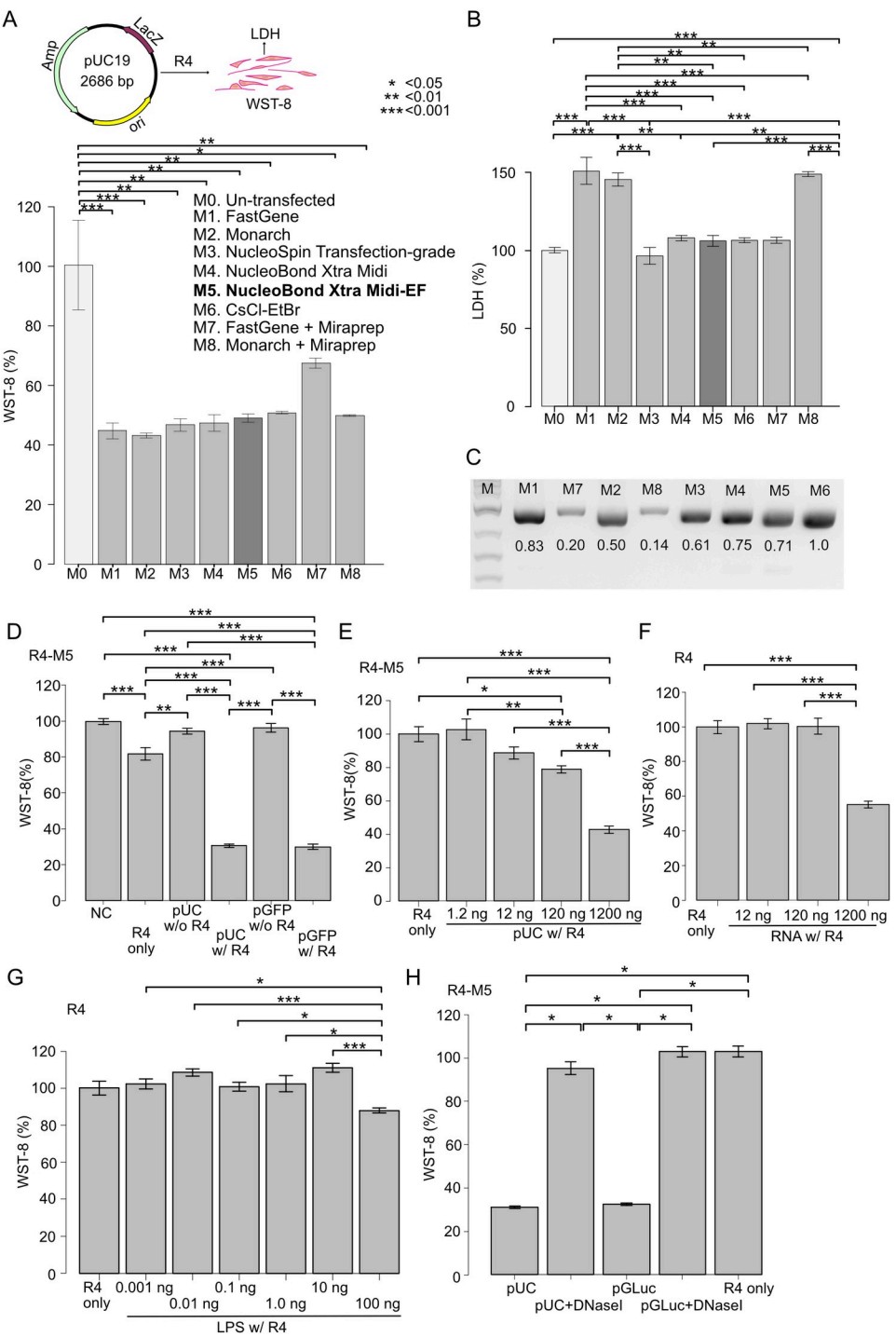

**Fig 2. Differences in cell viability by plasmid purification methods.** (A-B) Blank plasmid pUC19 was prepared by five commercially available mini- and Midi-kits, CsCl-EtBr centrifugation, and Miraprep protocol. Cell viability (WST-8) and cytotoxicity (LDH) were measured after transduction into cardiac fibroblasts. Bar graphs are the means of the biological duplicates and error bars showing the standard error of the mean. (A) pUC19 is a basic cloning vector, and does not express proteins by gene transfer. pUC19 samples were prepared by 8 different purification methods (M1-M8) and were introduced into HCF with GenomONE-GX (R4) similarly to the data represented in Fig 1. WST-8 were measured 48 hours later. Methods and kits are listed in descending order of expected sample purity, except for the Miraprep samples. As for the purity of Miraprep samples, it depends on the kits applied, so the bar graphs are separated from the other samples. The un-transfected condition means HCF in which neither reagents nor plasmids were added to the medium. (B) Measurement of supernatant LDH of HCF cells 48 hours after gene transfer. The

medium was changed once at 24 h after the transfection, and LDH was measured in the supernatants collected after another 24 h. (C) Agarose-gel electrophoresis of 1 µg pUC19 plasmids used in Fig 2A and 2B. Their plasmid concentrations were measured by Nanodrop, and the same amount of pUC19 samples were digested by unique restriction enzyme and run on an agarose-gel. There is a large discrepancy between the values by Nanodrop and actual ones in electroporesis. (D-H) Cell viability was measured in HCF transfected with pUC19 and GFP-expressing plasmids (D-E), E.coli RNA (F), and E.coli LPS (G), and GLuc-expressing plasmid (H). Bar graphs are the means of the biological replicates (n = 5) for (D-G) and (n = 6) for (H). Error bars represent the standard error of the mean. Statistical analysis was performed by one-way ANOVA & post-hoc Tukey (A-G), and Kruskal-Wallis test & post-hoc Steel Dwass (H). Asterisks represent different p-values calculated in the respective statistical tests (*: $p < 0.05$; **: $p < 0.01$; ***: $p < 0.001$).

**Polyethylenimine Max (Polysciences; Pennsylvania, USA) in Fig 1.** Polyethylenimine powder was dissolved in distilled water while dropping hydrochloric acid (final concentration 1.0 mg/mL), and adjusted in pH to 6.50 with hydrochloric acid or sodium hydroxide. Finally, it was sterilized through 0.22 µm PES filter, and stored at -30˚C till use. The PEI solution was thawed on the day of transfection, and put on a safety cabinet to warm it to room temperature before use. Once thawed, the PEI solution was stored at 4˚C, and repeatedly used for up to three months. After thawing and adjusting it to room temperature, PEI 1.0 mg/mL 8 uL was added to 27 µl OptiMEM. 1300 ng pLV-CMV-SNAPtag-DasherGFP was added to 35 µl Opti-MEM. PEI/OptiMEM was added to Plasmid/OptiMEM while vortexing gently, spinned down, incubated at room temperature for 20 minutes, then 35 µl plasmid/PEI/OptiMEM mixture was equally applied to two wells in a 24-well plate. Twenty-four hours later, the medium was removed and exchanged with fresh complete medium.

**Viofectin (Viogene; Taiwan, ROC) in Fig 1.** Viofectin reagent was stored at 4˚C, and put in a safety cabinet to warm it to room temperature and vortex shortly just before use. Viofectin 2 uL was mixed well with OptiMEM 60 uL, and incubated at room temperature for 5 minutes. 6 µg pLV-CMV-SNAPtag-DasherGFP was added to the Viofectin/OptiMEM, incubated at room temperature for 30 minutes, then Plasmid/Viofectin/OptiMEM 30 uL was equally

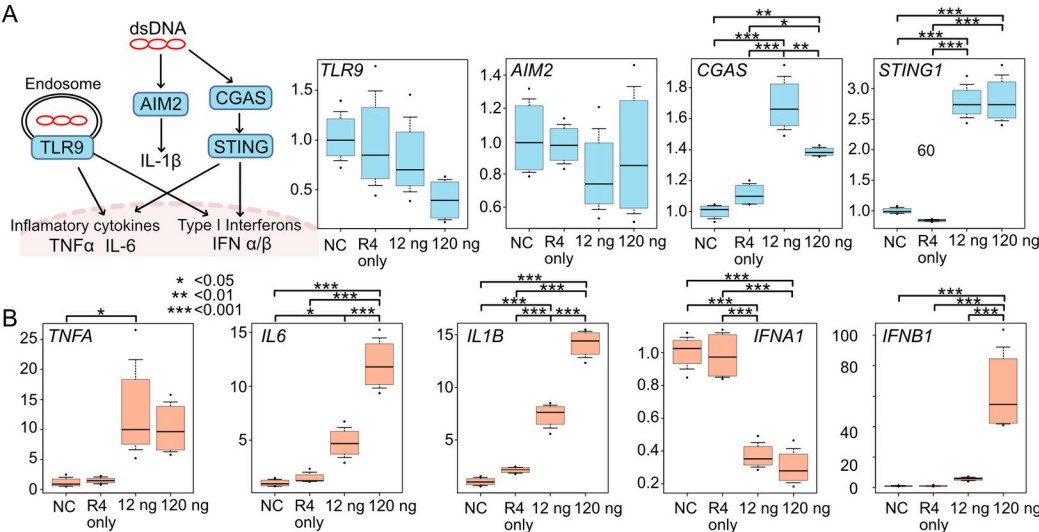

**Fig 3. Plasmid transfection activates dsDNA receptors and downstream cytokines.** (A-B). RT-qPCR analysis of double-stranded DNA receptors (TLR9, AIM2, CGAS, and STING1) and inflammatory cytokines (TNFA, IL6, IL1B, IFNA1, and IFNB1). Blank plasmid pUC19 was transfected into HCF. HCF transfected by only reagent R4 without plasmid acted as negative controls, and its values were set as 1.0. Box plots represent median, 10th, and 90th percentiles from the biological quadruplicates, except for that of TNFA, which is triplicate. Statistical analysis was performed by one-way ANOVA & post-hoc Tukey. Asterisks represent different p-values (*: $p < 0.05$; **: $p < 0.01$; ***: $p < 0.001$).

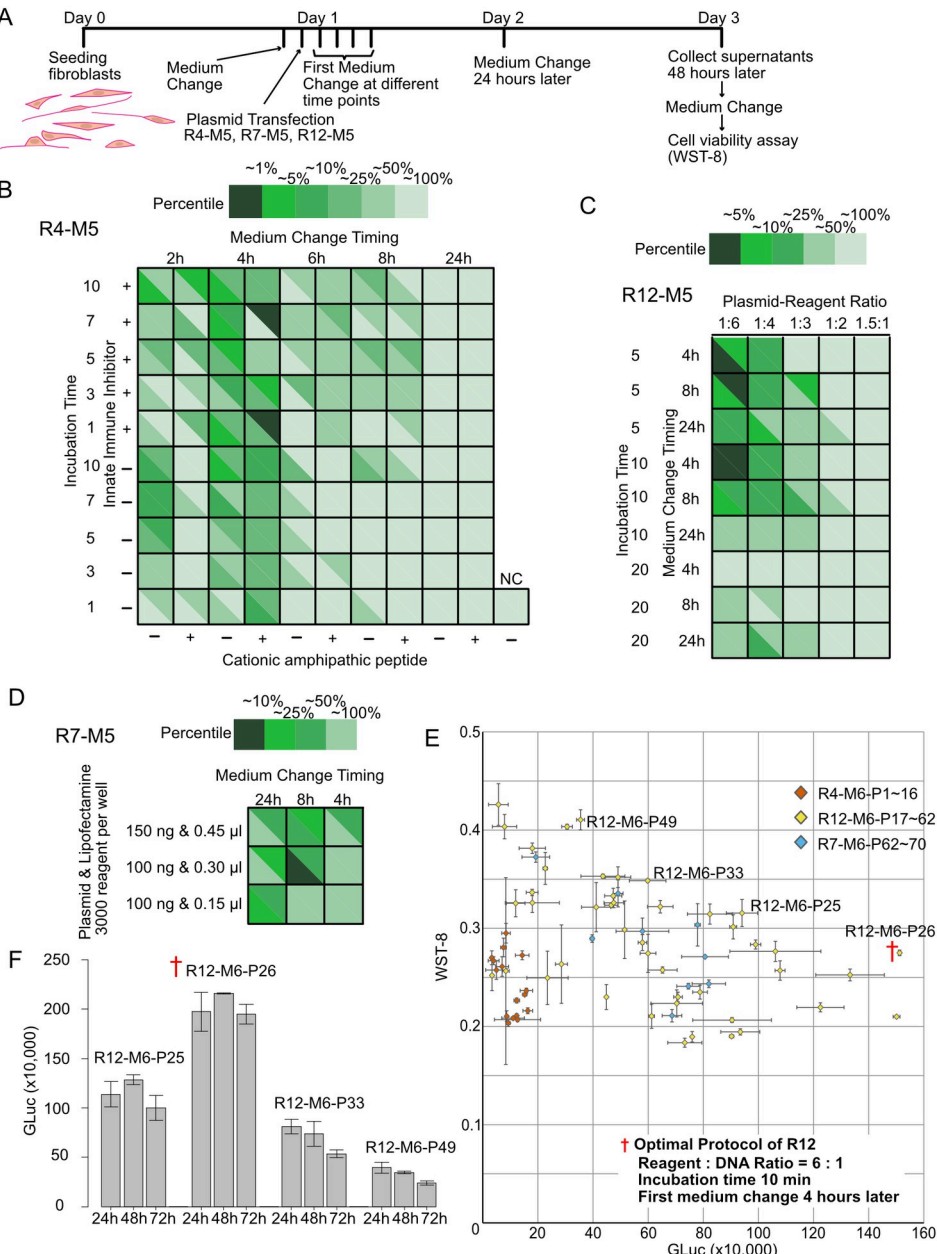

**Fig 4. Optimization of plasmid transfection protocols.** (A) Schedule of optimization experiments. The cells were seeded on day 0 and perform the gene transfer was performed on day 1. Medium replacement was done at 2, 4, 6, 8, or 24 h. At 24 h, the media of all conditions was replaced, and the amount of luciferase secreted into the medium during 24 h from Day 2 to Day 3 was compared between different conditions. CCK-8 assay was performed on Day 3. (B) Luciferase activity after transfection into HCF cells with Reagent #4 (R4) under 100 unique conditions. The variable parameters in R4 are the addition of KALA (cationic amphipatic) peptide or not, the addition of an inhibitor of innate immunity or not, and the incubation time from mixing reagent/plasmid to applying to the cells, and the timing of first medium change. Each condition was performed twice, and one square corresponds to one condition, divided by a diagonal line to show the repeated results. The luciferase activity is divided into top 1%, 5%, 10%, 25%, and 50% percentile, with the higher activity indicated by a darker color. (C) Luciferase activity after transfection into HCF cells with Reagent #8 under 50 conditions; variable parameters in #8 are the ratio of reagent to plasmid, the total amount of reagent and plasmid into cells, and the timing of first medium change. (D) Luciferase activity after transfection into HCF cells with Reagent #13 under 9 conditions; the variable parameters in #13 are the total amount of reagent and plasmid into cells, and the timing of the first media change. (E) Scatterplot of cell viability vs. luciferase activity, where CCK-8 was measured and plotted for those with high activity in Fig 4B, and all in 4C-4D. The details of the conditions marked with a red † are shown in the lower right corner of the scatter plot. Dots and error bars represent mean and standard error of the mean. (F) Time-series analysis of luciferase activities from three protocols selected from 4E.

applied to two wells in a 24-well plate. Twenty-four hours later, the medium was removed and exchanged with fresh complete medium.

**Viafect (Promega; Wisconsin, USA) in Fig 1.** Viafect reagent was stored at 4˚C, and put on a safety cabinet to warm it to room temperature and vortexed shortly just before use.

1000 ng pLV-CMV-SNAPtag-DasherGFP was added to 100 µl OptiMEM. 3 µl Viafect reagent was mixed with the Plasmid/OptiMEM, incubated at room temperature for 20 minutes, then 50 µl plasmid/Viafect/OptiMEM was equally applied to two wells in a 24-well plate. Twenty-four hours later, the medium was removed and exchanged with fresh complete medium.

All combinations of variable parameters (plasmid to reagent ratio [1:6/1:4/1:3/1:2/1.5:1], incubation time [5/10/20 minutes] and first medium change timing [4/8/24 hours after the transfection]) were tested in Fig 4.

**Avalanche-Omni Transfection Reagent (EZ Biosystems; Maryland, USA) in Fig 1.** Avalanche-Omni reagent was stored at 4˚C degree, and put in a safety cabinet to warm it to room temperature and vortexed shortly just before use. 4 µg pLV-CMV-SNAPtag-DasherGFP was added to 100 µl OptiMEM. 3.2 µl Avalanche-Omni reagent was mixed well with the Plasmid/OptiMEM, vortexed for 2 or 3 seconds incubated at room temperature for 15 minutes, then 50 µl plasmid/Viafect/OptiMEM was equally applied to two wells in a 24-well plate. Twenty-four hours later, the medium was removed and exchanged with the fresh complete medium.

**ScreenFectA (Fujifilm-Wako) in Fig 1.** ScreenFectA reagent and the Dilution buffer were stored at 4˚C, put in a safety cabinet to warm them to room temperature and vortexed shortly just before use. 2.5 µl ScreenFectA reagent was added to the 50 µl Dilution buffer. 1000 ng pLV-CMV-SNAPtag-DasherGFP was added to the 50 µl Dilution buffer. The ScreenfectA/ Dilution buffer was mixed well with the Plasmid/Dilution buffer, incubated at room temperature for 15 minutes, then 50 µl plasmid/ScreenFectA/Dilution buffer was equally applied to two wells in a 24-well plate. Twenty-four hours later, the medium was removed and exchanged with the fresh complete medium.

**jetOPTIMUS (Polyplus; Illkirch, France) in Fig 1.** jetOPTIMUS reagent and buffer were stored at 4˚C. 1000 ng pLV-CMV-SNAPtag-DasherGFP was added to jetOPTIMUS buffer 100 uL, vortexed for 1 second, spinned down. jetOPTIMUS reagent 1.0 uL was mixed with the Plasmid/jetOPTIMUS buffer, vortexed for 1 second, spun down, incubated at room temperature for 10 minutes, then Plasmid/jetOPTIMUS reagent/jetOPTIMUS buffer 50 uL was equally applied to two wells in a 24-well plate. Twenty-four hours later, the medium was removed and exchanged with the fresh complete medium.

**Helix-IN transfection reagent (OZ Biosciences; Marseille, France) in Fig 1.** Helix-IN reagent and HIB100X were stored at -20˚C, and put in a safety cabinet to warm to room temperature. 1.5 µg pLV-CMV-SNAPtag-DasherGFP was added to 100 µl OptiMEM, 2.24 µl Helix-IN reagent was mixed with the Plasmid/OptiMEM, incubated at room temperature for 30 minutes, then 50 µl plasmid/Helix-IN/OptiMEM mixture was equally applied to two wells in a 24-well plate, and 5 µl HIB100X was applied to each transfected well. Twenty-four hours later, the medium was removed and exchanged with fresh complete medium.

**GenomONE-GX (distributed by Fujifilm-Wako insted of Ishihara sangyo; Osaka, Japan) in Fig 1.** GenomONE-GX protocol consists of Reagent 1, 2, 3, 4, 5, enhancer solution, and KALA peptide. Reagent 3 is an inactivated Hemagglutinating Virus of Japan (referred to as HVJ-E). Enhancer solution inhibits the innate immunity against exogenous double-strand DNAs. KALA peptide solution helps the plasmid's endosomal escape, and was separately purchased from Fujifilm-Wako. Reagent 3 was vortexed for 2 to 3 seconds before use. All reagents were stored at 4˚C, and kept on ice during the procedure.

In Fig 2, Reagent 1 1.7 μl was collected in a PCR tube on ice. 10 μl Reagent 2 was mixed well with the Reagent 1. pLV-CMV-SNAPtag-DasherGFP were diluted with Tris-EDTA buffer to 200 ng/uL. The 10 μl diluted plasmid was mixed well with the Reagent 1/2. 10 μl Reagent 3, inactivated HVJ-E, was mixed well with the Reagent 1/2/plasmid. 10 μl KALA peptide solution 0.02 mg/mL was mixed well with the Reagent 1/2/plasmid/3. 10 μl Reagent 4 was mixed well with the Reagent 1/2/plasmid/3/KALA, incubated on ice for 5 minutes, then 25 μl Reagent 1/2/plasmid/3/KALA/4 mixture was applied to two wells in a 24-well plate, and the 12.5 μl enhancer solution was applied to each transfected wells. Twenty-four hours later, the medium was removed and exchanged with the fresh complete medium.

GenomONE-GX (Reagent 5 + KALA) protocol used Reagent 5 instead of Reagent 3 in Fig 1. And this GenomONE-GX (Reagent 5 + KALA) protocol was optimized further. All combinations of variable parameters (incubation time [1/3/5/7/10 minutes], use or non-use of KALA peptide and enhancer, first medium change timing [2/4/6/24 hours after the transfection]) were tested in Fig 4.

**PolyMag Neo magnetofection reagent (OZ Biosciences) in Fig 1.** PolyMag Neo is a magnetofection reagent composed of magnetic nanoparticles coated with cationic molecules. PolyMag Neo reagent was stored at 4 Celsius degree, and vortexed shortly just before use. pLV-CMV-SNAPtag-DasherGFP 2000 ng was added to 200 μl OptiMEM. The Plasmid/OptiMEM was mixed well with 2 μl PolyMag Neo solution, incubated at room temperature for 30 minutes, then the 33 μl plasmid/PolyMag Neo/OptiMEM mixture was applied in a dropwise fashion to two wells in a 24-well plate. The magnetic plate was put under the plate and incubated for 30 minutes in an CO2 incubator. Twenty-four hours later, the medium was removed and exchanged with the fresh complete medium.

**Lentivirus production.** Lentivirus production was performed as previously described [22] and based on Addgene's official protocol (https://www.addgene.org/protocols/lentivirus-production/), with some modifications. 10-cm dishes were coated with Collagen type I (Nippi; Tokyo, Japan) to increase the attachment of Lenti-X 293T cells and enable us to transfect lentiviral plasmids into 293T cells on the same day of seeding. Four 10-cm dishes were coated with Collagen Type I for 3 hours at room temperature (Day 0). After washing twice with Phosphate buffered saline without magnesium and calcium (PBS(-), Nacalai-tesque), 1.0e7 Lenti-X 293T cells were seeded on the coated-dishes and incubated in a $CO_2$-incubator for 6 hours. 40 μg pLV-CMV-SNAPtag-DasherGFP and 40 μg helper plasmid mixture (3rd Generation Packaging System Mix, LV053, Applied biological Materials, USA) were diluted with OptiMEM up to a total volume of 500 μl. PEI Max pH6.50 1.0 mg/mL was diluted with OptiMEM up to a total volume of 500 μl. The PEI/OptiMEM was added drop-wise into the plasmid/OptiMEM, while vortexing, and incubated at room temperature for 15 minutes. The 250 μl plasmid/PEI/OptiMEM mixture was applied to Lenti-X 293T cells. The medium was exchanged 18 hours after the transfection (Day 1). On day 2, 3, and 4, the supernatants were collected, filtered through 0.45 um PVDF membrane (MilliporeSigma; Missouri, USA), and centrifuged at 1000 g for 10 minutes. The cleared supernatants were collected and mixed with 3 volumes of a concentrator solution (40%w/v PEG8000, 1.2M NaCl, pH 7.2). The supernatant/PEG mixtures were gently shaken at 60 rpm in a cold room till Day 4. On day 4, the supernatant/PEG solutions that was previously collected on days 2–4, were pooled and centrifuged with a swing out rotor at 1600 g for 60 minutes at 4˚C degree. The viral vector pellets were resuspended with 4 mL PBS (-), and 500 μl aliquots in cryotubes were snap frozen in liquid nitrogen, and stored at -160˚C. The titer was measured with Lenti-X GoStix Plus (Takara-Bio).

**Lentiviral transduction.** Frozen lentiviral vector aliquot LV-CMV-SNAPtag-DasherGFP was thawed on ice. After PEG precipitation was dissolved by pipetting, 25 μl viral vector was applied into each 24-well plate.

**Image acquisition and analysis.** All cell images were taken by an inverted microscope (ECLIPSE Ts2, Nikon; Tokyo, Japan). Exposure time was 500 ms and the acquisition parameters were fixed in Fig 1. Gel image was acquired by ChemiDoc Imaging systems (Bio-Rad Laboratories; California, USA), and analyzed by Fiji, a package of ImageJ 1.53q.

## Luminescent assay

Coelenterazine powder 1 mg (BLD Pharmatech; Shanghai, China) was dissolved in 1 mL 99.5% ethanol and diluted by 100 mL PBS(-). 10 µl Tween20 (MilliporeSigma) and 20 µl EDTA (Invitrogen; California, USA) were also added. The final GLuc substrate [Coelenterazine 10 µg/mL, 0.01% v/v Tween20, 0.1mM EDTA in PBS(-)] was split into a cryovial and stored at -80˚C. On the day of the luminescent assay, cell supernatants and GLuc substrate, both stored at -80˚C, were thawed, 20 µl supernatant and 80 µl substrate were mixed together and luminescence was measured using NIVO (PerkinElmer; Massachusetts, USA). CLuc luminescence, used as a control in Fig 5, was detected by mixing 20 µl supernatant and 80 µl substrate (Pierce Cypridina Luciferase Glow Assay Kit, Thermo Scientific) diluted 2/5 in PBS(-).

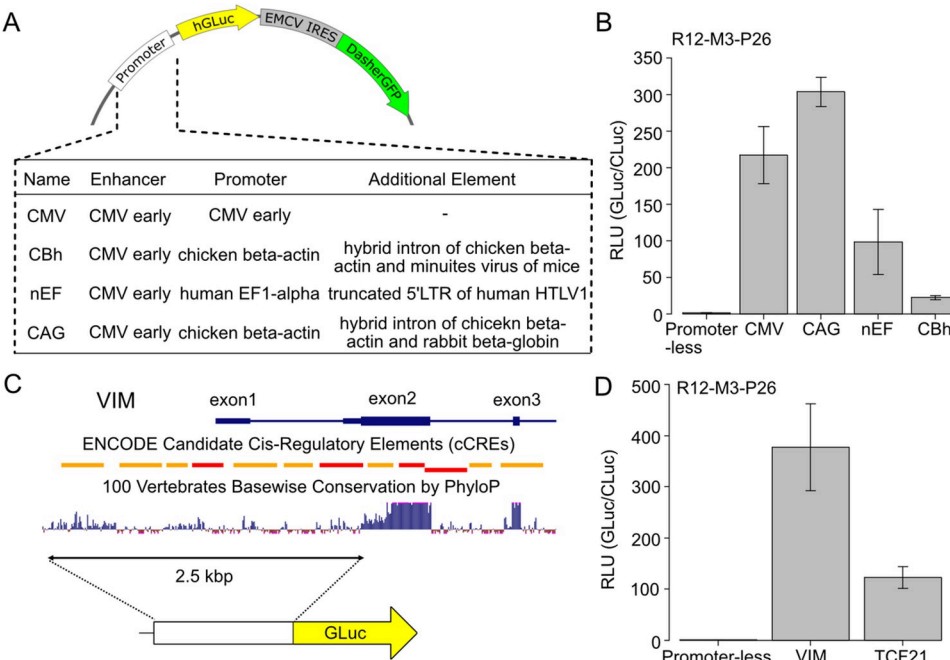

**Fig 5. Promoter assay with optimized protocol for primary human cardiac fibroblasts.** (A) Map of the plasmids used in the ubiquitous promoter assay. The secreted luciferase GLuc and the green fluorescent protein DasherGFP linked by IRES sequences are expressed downstream of the indicated promoters. In addition, WPRE, a post-transcriptional regulatory element, and a synthetic polyA signal are inserted downstream of the GFP for stabilization of the expressed mRNA. (B) Comparison of the viral and synthetic promoters' activities in HCF. Each plasmid with different promoters was transfected into HCF. GLuc activity in the medium was measured 48 hours after the transduction. All experiments were performed in quadruplicates, bars represent means and error bars are standard error of the mean. (C) Candidate isolation of predicted endogenous enhancer-promoter sequences. The dark blue horizontal bar represents exons and introns of human Vim mRNA (NM_003380.5). The histogram represents the basewise conservation of 100 vertebrates analyzed by PhyloP. In ENCODE Candidate Cis-Regulatory Elements (cCREs), Red boxes indicate promoter-like signature and orange boxes proximal enhancer-like signature. All data were extracted from USCS Genome Browser. The 2.5 kbp covering 5'UTR proximal enhancers-exon1-intron1-exon2-translation start site (TlSS) was amplified and put just upstream the GLuc gene on a plasmid. (D) Promoter/Enhancer activities of endogenous genes characteristic of fibroblasts (VIM and TCF21). GLuc plasmid and control plasmid constitutively expressing CLuc were transduced into HCF. Luciferase activity in the medium was measured 48 hours after the transduction. Luminescence of GLuc was normalized to the luminescence of the reference CLuc (GLuc/CLuc), then the relative values with Promoter-less as 1.0 are shown. All experiments were performed in quadruplicates, bars represent means, error bars are standard error of the mean.

## CCK-8/WST-8 assay

Cell proliferation assay was performed with formazan-based colormetric Cell Counting Kit-8 (Dojindo; Kumamoto, Japan) to check the cytotoxicity of plasmid transfection in Figs 2 and 4. 10 uL reagent was added into each well in a 96-well plate, and 450 nm absorbance was measured by NIVO.

## LDH assay

Released lactate dehydrogenase from damaged cells were detected with Cytotoxicity LDH Assay Kit-WST (Dojindo). Following the official protocol, orange Formazan's absorbance at 490 nm was measured by NIVO.

## RT-qPCR analysis

HCF cells were plated on a 96-well plate, and pUC19 plasmid was transfected by GenomONE-GX. Fourty-eight hours later, CCK-8 assay was performed, and direct lysis and cDNA conversion of HCF were conducted with SuperPrep II Cell Lysis & RT Kit for qPCR (Toyobo). mRNA copy number was measured with THUNDERBIRD Next SYBR qPCR Mix (Toyobo). The following qPCR primer pairs against dsDNA receptors (TLR9, CGAS, STING, and AIM2), pro-inflammatory cytokines (TNFA, IL6, IL1B, IFNA1, and IFNB1), and internal controls (HPRT1, and GUSB) were used (Table 1). Relative expression levels were normalized by multiple internal controls and calculated by $\Delta\Delta$-Ct method. Non-transfected condition was set as 1.

## Statistical analysis

For statistical analysis, the data was evaluated for significance by performing one-way ANOVA using EZR version 1.55, a package of R version 4.1.3. X-Y scatterplot with error bars in Fig 4D was depicted by Excel (Microsoft; Washington, USA). P values $< 0.05$ were considered to represent statistically significant differences. All data are presented as means and the standard error of the mean.

# Results

## Chemical transfection into primary human cardiac fibroblasts

The general outline of this study is depicted in Fig 1A. We procured primary human cardiac fibroblasts (HCFs) and expanded them to passage 8–9, after which we used the cells for transfection experiments with efficacy and cytoxicity assessments. To develop a suitable protocol

**Table 1. qPCR primer sequences.**

| Gene name | Forward primer | Reverse primer |
|---|---|---|
| TLR9 | ACGGCATCTTCTTCCGCTCA | AGGGAAGGCCCTGAAGATGC |
| CGAS | GACCACCTGCTGCTCAGACT | GTAGCTCCCGGTGTTCAGCA |
| STING1 | ATTGGACTGTGGGGTGCCTG | CGGTCTGCTGGGGCAGTTTA |
| AIM2 | GGTTATTTGGGCATGCTCTCCTG | AAACCGGGTCTGCCACCTTC |
| TNFA | CCTGCTGCACTTTGGAGTGA | GTCACTCGGGGTTCGAGAAGA |
| IL6 | AGCCCACCGGGAACGAAAGA | GAAGGCAACTGGACCGAAGG |
| IL1B | TCTTCGAGGCACAAGGCACA | TTTCACTGGCGAGCTCAGGT |
| IFNB1 | CCTGTGGCAATTGAATGGGAGG | AGATGGTCAATGCGGCGTCC |
| IFNA1 | TTCAAAGACTCTCACCCCTGC | ACAGTGTAAAGGTGCACATGACG |
| HPRT1 (house keeping) | AGCCCTGGCGTCGTGATTAG | TCGAGCAAGACGTTCAGTCCTG |
| GUSB (house keeping) | CGTCTGCGGCATTTTGTCGG | ACCCCATTCACCCACACGA |

for nucleic acid transfection of HCFs, we conducted an initial screening of 12 commercially available chemical transfection reagents (R2 to R13) (Fig 1B).

Quantitative fluorescence intensity analysis following transfection with a GFP-encoding plasmid revealed efficiencies of 2.28% for R2, 17.65% for R3, 21.7% for R4, 1.55% for R6, 3.08% for R7, 2.77% for R12, and 4.95% for R13 relative to lentiviral vector-transduced positive control (R1) (Fig 1C). All other reagents yielded less than 1% efficiency.

Assessment of dead cells by propidium iodide (PI) demonstrated 14.05% PI signal for R7, 23.47% for R8, 10.64% for R10, and 13.53% for R12, compared to the highest R3, while other reagents induced over 32% PI fluorescence (Fig 1D).

Based on these initial results, reagents R7 and R12 were selected as lead candidates for further optimization (Fig 1B–1D). Additionally, R4 was retained as a potential optimization candidate despite reduced viable cells, owing to the highest transfection efficiency among R2-R13.

## Effect of plasmid purification methods on fibroblast viability

As high plasmid purity is vital for efficient gene delivery into primary cells [23–25], we evaluated the impacts of various purification methods (M1 to M8) on HCF viability following R4-mediated plasmid transfection.

Analysis showed a ~50% reduction of viability for groups transfected with M1-M8-purified plasmids versus non-transfected control (M0), albeit slightly higher survival rates for M7 (Fig 2A). Lactate dehydrogenase (LDH) release assessment revealed greater cytotoxicity from M1, M2, and M8 versus other techniques (Fig 2B). Notably, gel electrophoresis indicated that M7 and M8 yielded just 20% and 14% desired product versus the maximum purity, centrifugation-based M6 method (Fig 2C).

While adding plasmid alone to the medium did not reduce viability (pGFP 96.5%; pUC19 cloning vector 94.6%) compared to untreated cells, adding R4 reagent without plasmid decreased survival to 82%. Further decline occurred following addition of R4-plasmid complexes, regardless of transgene expression (pGFP 30.0%; pUC19 30.6%) (Fig 2D). As a next step, we evaluated if the concentration of plasmid affected the cell viability. We found that the viability decreased proportionally as transfected plasmid amounts increased at constant R4 concentration (Fig 2E). Additionally, supplementing R4 with high quantities of either E. coli RNA or lipopolysaccharides (LPS) revealed no cytotoxicity except for the highest quantity (RNA 120 ng 100.4%; LPS 10 ng 110.9% in Fig 2F and 2G). Finally, DNase I treatment demonstrated a recovery of survival rates to levels comparable to R4-only condition (pUC19, 30.2% before; 92.4% after; pGLuc, 31.4% before; 99.9% after) (Fig 2H).

Taken together, these findings suggest that nucleic acids affect the cell viability more than impurities. As a next step, we examined typical pathways underlying nucleic acid introduction to cell death. This analysis showed an upregulation of inflammatory cytokines, such as TNFA (8.5-fold), IL6 (11.6-fold), IL1B (13.3-fold), and IFNB1 (61.3-fold) compared to non-transfected cells (Fig 3A), along with an increase in CGAS (1.38-fold), a double-stranded DNA receptor, and its downstream mediator STING (2.82-fold) (Fig 3B).

Based on these findings, among the comparable plasmid purification methods (M3-M6), we selected the moderate-scale (~ 200 mL) M5 approach in our optimization steps. For high-throughput (~ 3 mL) experiments, we chose the small-scale amenable M3, which is the most convenient approach.

## Exploration of parameters affecting gene transfer into fibroblasts

With limited enhancements from optimizing plasmid purification alone, we adopted a systematic strategy to evaluate numerous parameters throughout the transfection protocol (Fig 4A

and S1 Table). A total of 100 conditions were assessed for R4, 45 for R12, and 9 for R7, spanning reagent-plasmid mixing approaches to post-transfection cell handling.

Notably, the timing of initial media changes strongly influenced gene delivery with R4, with the highest luciferase activity observed when changing media at 4 hours post-transfection and the lowest at 24 hours (Fig 4B). For R12, luminescence rose with higher R7 volumes for the same plasmid amount (peak at 0.6 μl for 100 ng plasmid) and optimal media change timing between 4 and 8 hours (Fig 4C). Maximum luminescence occurred with R7 under 100 ng plasmid and 0.30 μl reagent at 8 hours post-transfection for media replacement (Fig 4D).

Subsequent plotting of luciferase against cell viability (Fig 4E) revealed an inverse correlation (Pearson's $r = -0.481$, $p = 0.0595$ for R4; $r = -0.538$, $p = 0.000136$ for R12; $r = -0.741$, $p = 0.0224$ for R7), indicating a trade-off between transfection efficiency and survival rate. Consequently, we selected one specific combination, termed "P26" (†), achieving the optimal balance of viability and gene delivery at 48 hours. Furthermore, P26 enabled sustained high-level expression over 24–72 hours (Fig 4F), validating utility for medium-term experiments.

## Promoter assay using human primary cardiac fibroblasts

In a final demonstration of our optimized gene delivery methodology, we conducted a plasmid-based promoter assay (Fig 5), featuring four ubiquitous promoters—CMV [26], CBh [27], nEF [28], and CAG [29]—as well as newly reconstructed regulatory regions governing classical fibroblast marker VIM [30] and resident fibroblast marker TCF21 [31,32].

Among the ubiquitous promoter activities, CMV, utilized previously for our GFP and luciferase expression experiments, exhibited the second highest signal at 216.9-fold above baseline, with CAG showing the maximum at 303.6-fold (Fig 5B). The nEF and CBh promoters, commonly used in various cell types, demonstrated modest increases of 98.1-fold and 22.0-fold, respectively.

Additionally, we examined the newly reconstructed VIM and TCF21 promoter/enhancer sequences (Figs 5C and S1). The VIM regulatory region yielded remarkable 350-fold activation over background, while TCF21 showed substantial 120-fold upregulation (Fig 5D). These activities persisted, albeit at lower levels, when switching the transfection reagent from R7 to R4 (82.6-fold VIM, 8-fold TCF21 change) or R12 (204-fold VIM, 37-fold TCF21 change) (S2 Fig).

Furthermore, when applying this gene delivery method to human fibroblasts derived from the lung, kidney, and skin, enhancer/promoter sequences of genes such as DDR2 and DCN, which are identified as markers in fibroblasts by scRNA-seq, exhibited high activity in all tested cell types (S3 Fig).

In summary, our results validate that this optimized transfection protocol can capture endogenous transcriptional activities in human primary fibroblasts with exceptional sensitivity. We propose this methodology as an indispensable approach for dissecting gene regulatory mechanisms governing fibroblast phenotype and function in cardiac disease.

## Discussion

Our study on plasmid introduction into primary human cardiac fibroblasts (HCF) is, to our knowledge, the first of its kind. The nuances of plasmid delivery into primary cells necessitate tailored optimization for each cell type, often resulting in numerous failed attempts. In this report, our success in achieving a delicate balance between cell viability and transfection efficiency through meticulous reagent and protocol optimization stands out.

Conventionally, viral vectors are extensively employed [17,18]. Our in-house lentiviral vector, consistent with previous reports, exhibited high expression levels with minimal cell death

(Fig 2). However, the labor-intensive nature of viral vector production led us to explore a simpler plasmid-based introduction method. Additionally, using plasmids circumvents potential interference from promoter/enhancer activities derived from viral sequences on a viral vector [33] and around the genome integration sites [34], thereby offering a clearer insight into the luciferase assay results on gene regulatory elements.

Focusing strictly on plasmid-based gene introduction, we demonstrated that plasmids purified from tube-scale E. coli cultures with transfection-grade kit (M3) show equivalent cell viability compared to the traditional flask-scale E. coli cultures with endotoxin-free kit (M5) or CsCl-EtBr centrifugation (M6) (Fig 2A and 2B). The dynamic range of luciferase assays following plasmid introduction exceeded 100–350 times the background levels, facilitating the simplification of enhancer and promoter sequences to identify core elements (Fig 5D).

Our method's versatility is highlighted by sustained gene expression for up to 72 hours (Fig 4F), making it suitable for medium-term reporter assays observing cellular responses to compounds. Indeed, we successfully introduced signaling pathway-specific reporter plasmids into HCF, added compounds to the culture medium after 24 hours, and performed measurements at 48 and 72 hours without complications (S4 Fig).

In our study, the seeding density for human cardiac fibroblasts (HCFs) during plasmid transfection was uniformly set at $1.82 \times 10^4$ cells/cm$^2$ across all conditions. This seeding density ensures that the HCFs utilized in our experiments reach confluence approximately 96 hours post-seeding under standard culture conditions without genetic manipulation. For luciferase reporter assays investigating signal cascades, plasmid transfection is typically performed 24 hours post-seeding, followed by stimulation at around 48 hours, and sample measurement at 48–72 hours post-seeding. Therefore, this seeding density was meticulously chosen to maintain cell proliferation without reaching confluence during the critical observation period.

HCFs exhibit density-dependent behaviors; at lower densities, they become rounded and proliferate slowly, while at higher densities, they elongate and cease division. Considering that transfection efficiency tends to be higher in actively dividing cells, it is crucial to maintain an optimal seeding density to ensure reproducibility and reliability of the results. Thus, our protocol optimization with Viafect (R12) achieved maximum and consistent luciferase activity between 48–96 hours post-seeding (24–72 hours post-transfection), making it suitable for most experimental purposes (Fig 4E). However, for experiments requiring different schedules, we recommend first adjusting the total amount of plasmid/reagent complex based on the specific seeding density and then preparing the timing of the first medium change between 2 and 6 hours post-transfection (S2 Table).

Conversely, the protocol optimized with Lipofectamine 3000 (R7) showed peak activity at 48 hours post-transfection, with a decline observed at 72 hours. The temporal profile of expression is closely related to the total amount of plasmid and Lipofectamine 3000 reagent added per well. Low amounts (plasmid:reagent = 100 ng:0.15 μl) achieved maximum expression at 48 hours, while higher amounts (plasmid:reagent = 150 ng:0.45 μl) peaked at 24 hours, with intermediate amounts (plasmid:reagent = 100 ng:0.30 μl) showing a middle-ground temporal profile (S5 Fig). Therefore, Lipofectamine 3000 serves as a viable alternative for assays requiring measurements at a single time point between 24–48 hours post-transfection.

Challenges in plasmid introduction into primary cells arise from significant variations in transfection efficiency (Fig 1A) and subsequent cell death (Fig 1B). While the precise composition of reagents impedes investigation into the former, we have been addressing the latter. Our initial hypothesis centered around the potential induction of inflammatory cytokines and apoptosis by overexpressed foreign proteins [35], E. coli-derived lipopolysaccharides (LPS) [36], RNA, or the plasmid itself [37]. Experiments presented in Figs 1 and 2 aimed to isolate these factors. First, robust expression of foreign proteins via lentiviral vectors did not induce

significant toxicity (Fig 1A). However, plasmid lacking protein expression (pUC19) led to cell death (Fig 2D), eliminating the dependency on foreign protein production for low cell viability. Subsequent experiments with LPS and RNA transfection did not increase cell death (Fig 2F and 2G). Furthermore, the survival rate recovered to negative control levels when plasmids were digested with DNase I while maintaining the composition of the plasmid solution (Fig 2H), suggesting that LPS and RNA were not the primary causes of decreased cell viability. These findings elucidate why plasmids purified by the Miraprep exhibited a high survival rate. The actual plasmid concentrations of the Miraprep product were strikingly reduced compared to that measured by Nanodrop (Fig 2C). Consequently, less amount of plasmid was introduced compared to the other groups, and the reduction in plasmid quantity increased survival (Fig 2E).

Attempts to enhance cell viability by improving plasmid quality failed, and our focus shifted to exploring the involvement of the TLR9/AIM2/cGAS-STING pathways and the potential triggering of innate immunity by plasmid DNA. RT-qPCR analysis revealed a significant upregulation of cytoplasmic DNA sensor cGAS, downstream STING (Fig 3A), and inflammatory cytokines (TNFA, IL1B, IL6, IFNB1) (Fig 3B) following plasmid transfection. Conversely, TLR9 and IFNA1 exhibited a decrease. Subsequent information from the supplier revealed that the additive attached to the reagent R4 acts as an innate immune inhibitor. While the target protein of the inhibitor was not disclosed by the supplier, we speculated that this could explain the observed effects on TLR9 and IFNA1. Activation of the cGAS-STING pathway in human cardiac fibroblasts remains unclear; however, evidence from studies on mouse embryonic fibroblasts [38] and patient-derived vulvar cancer-associated fibroblasts [39] suggests its presence.

Unfortunately, the expression of fluorescent proteins proved challenging. While reagent R4 demonstrated clear fluorescence (Fig 1B), abnormal cell morphology and high cell death were observed in 48 hours. Although our optimized protocol minimized these issues, fluorescence expression was compromised. Specifically, the transfection efficiency, as indicated by the expression of fluorescent proteins, was less than 5% for both R7 and R12. The inherent difficulties arise from the fact that a plasmid, not the impurities, induces these abnormalities and cell death. As a strategy to mitigate this plasmid-induced cell death, inhibitors interfering with DNA sensors have been identified [40]. However, their use could alter fibroblast physiology, potentially distorting experimental results. Their application might be carefully selected based on experimental goals. For example, in genome editing where the aim is to knock-in genes, treating cells with an inhibitor cocktail along with donor DNA introduction enhances cell survival and knock-in efficiency. Yet, for many experiments utilizing primary cells, the goal is to observe their physiological functions. In such cases, conventional approaches like lentiviral vectors or mRNA introduction, renowned for achieving high expression across various cell types, may be the only practical choices to balance the expression of fluorescent proteins with preservation of normal cellular states.

Looking ahead, we aspire to leverage our novel methodology to establish fibroblast-specific gene control. While observational studies, such as single-cell sequencing, have significantly advanced our understanding of fibroblasts over the past decade, interventions, particularly in the realm of genetic manipulation, have seen limited progress. In the cardiovascular field, research has predominantly focused on cardiomyocytes rather than cardiac fibroblasts, and tool development has been vigorously pursued in the former. Indeed, various tools enabling specific control *in vivo* are available for cardiomyocytes, such as recombinant AAVs (rAAV2/6, 2/7, 2/8, 2/9) [41], cTnT [42], and MHC [43] promoters. In contrast, cardiac fibroblasts have only a few promoter [17] and transgenic animals [44] available. Leveraging our newly developed method, we envision identifying a variety of cardiac fibroblast subtype-specific

promoter and enhancer sequences (S3 Fig). This endeavor will open up new avenues for functional analysis of heterogeneous fibroblasts *in vivo* by genetic tools.

## Supporting information

**S1 Fig. Engineering synthetic VIM and TCF21 enhancer/promoter plasmids.** The human Vim (A) and Tcf21 (B) genomic locus with H3K27Ac Mark on seven cell lines from ENCODE, alignments of 100 vertebrates, and ENCODE Candidate Cis-Regulatory Elements (cCREs) combined from all cell types. cCREs are labeled according to the regulatory signatures: Red box, promoter-like signature; orange box, proximal enhancer-like signature; yellow box, distal enhancer-like signature. All data was extracted from USCS Genome Browser. (A) Construction strategy of pGL4.14(Vim, GLuc-WPRE-SpA). Two Vim mRNA sequenes (ENST00000544301.7 and ENST00000224237.9) are registered and there are two promoter-like signatures (red box) in cCREs. PCR 2.5 kbp including both promoter-like signatures, first exon and intron of ENST00000544301.7, second exon of ENST00000544301.7 (this is a first exon of ENST00000224237.9), and the annotated translation start site (TISS), then insert into Eco31I site of a pGL4.14 reporter vector. The secreted luciferase GLuc is expressed downstream of the inserted enhancer/promoter candidates. In addition, synthetic polyadenylation signal and RNA polymerase II transcriptional pause signal from the human α2 globin gene are upstream of the candidate sequences to suppress transcriptions from the vector backbone Amp and ori. (B) Construction strategy of pGL4.14(Tcf21, GLuc-WPRE-SpA). Two Tcf21 mRNAs (ENST00000367882.5 and ENST00000237316.3) are registered and two antisense RNA (TARID, ENST00000607033.5 and ENST00000630119.2) also transcribe from the same locus. There are two promoter-like signatures (red box) in cCREs, but one signature (dashed box) may be associated with the antisenses. PCR 0.9 kbp enhancer and Tcf21 promoter-like signiture, 1.6 kbp intragenic enhancer signatures, and the annotated TISS, then insert 0.9 kbp into Eco31I site and 1.6 kbp into BglII site of the pGL4.14.
(TIF)

**S2 Fig. Promoter/Enhancer activities of the endogenous genes (VIM and TCF21) in HCF transfected by Reagent #4 or #7.** GLuc plasmid and control plasmid constitutively expressing CLuc were transduced into HCF. Luciferase activity in the medium was measured 48 hours after the transduction. Luminescence of GLuc was normalized to the luminescence of the reference CLuc (GLuc/CLuc), then the relative values with Promoter-less as 1.0 are shown. All experiments were performed in quadruplicate, bars are means, error bars are standard error of the mean.
(TIF)

**S3 Fig. Cell viability and endogenous gene promoter/enhancer activity in primary human fibroblasts derived from lung (A), kidney (B), skin (C), and auricle of heart (D).** Transfection reagent was Reagent #12, plasmids were prepared by Method #3, and the transfection protocol was Protocol #26 (R12-M3-P26; the optimal protocol in HCF). Each plasmid encodes GLuc under the control of a different enhancer/promoter sequence. Cell viability (WST-8) and luciferase activity in the medium were measured 48 hours after the transduction. Relative luminescence values are shown with promoter-less as 1.0. All experiments were performed in triplicate, bars are means, error bars are standard error of the mean.
(TIF)

**S4 Fig. JAK/STAT signal cascade analysis in HCF cells using optimized protocol with Reagent #12.** Luciferase assay measuring the activation of the JAK/STAT signal cascade in human cardiac fibroblasts (HCF) transfected with an optimized protocol using reagent 12.

The plasmid contains GLuc under the control of two STAT6 binding sites upstream of a TATA box, allowing GLuc expression upon STAT6 nuclear translocation. Cytokine IL-4 was added during the initial medium change, and supernatants were collected 48 hours post-transfection for luciferase activity measurement. Relative luminescence values are shown with unstimulated as 1.0. All experiments were performed in triplicate, bars are means, error bars are standard error of the mean.
(TIFF)

**S5 Fig. Time-series analysis of GLuc activities in HCF transfected by Reagent #7, related to Fig 4F.** GLuc plasmid was transduced into HCF. Luciferase activity in the medium was measured 24, 48, and 72 hours after the transduction. Luminescence of GLuc was measured. All experiments were performed in duplicate, bars are means, error bars are standard error of the mean.
(TIFF)

**S1 Table. Details of the protocols listed in the scattor-plot in Fig 4E.** Luciferase luminescence and WST-8 values were measured by duplicate.
(XLSX)

**S2 Table. Optimal Transfection Protocols of GenomONE-GX (R4), Lipofectamine 3000 (R7), and Viafect (R12) related to Figs 5 and S2.** All plasmids used are either transfection-grade or prepared by midiprep.
(DOCX)

**S1 File. FASTA format file of pGL4.14(TCF21pro, GLuc-WPRE-TCF21_E-SpA) in Fig 5D.** One potential enhancer region and one potential promoter region upstream of the human TCF21 gene were inserted upstream of GLuc, and three potential enhancer regions downstream of the human TCF21 gene were inserted downstream of WPRE.
(FA)

**S2 File. FASTA format file of pGL4.14(VIMpro, GLuc-WPRE) in Fig 5D.** Five potential enhancer regions and two potential promoter regions upstream of the human VIM gene were inserted upstream of GLuc.
(FA)

**S3 File. Raw data of the bar graphs in Figs 1–5 and S2, S3 and S5 Figs.** Luciferase luminescence and WST-8 values in the corresponding protocols are listed.
(XLSX)

**S1 Raw images. Original image file of agarose gel in Fig 2C.** The lanes to the left of the molecular weight marker (M) contain plasmids purified by each method, loaded at a concentration of 1000 ng/lane as measured by Nanodrop. The lanes to the right contain the plasmids after a single restriction enzyme digestion. The dotted lines indicate the outline of the agarose gel.
(PDF)

## Acknowledgments

We thank Dr. Bertrand-David Segard (Metcela Inc.) for his advice about cell culture assay, Dr. Sebastian Sjoeqvist (Metcela Inc.) for his comments on the manuscript, and Ms. IMAMURA Tomomi (Metcela Inc.) for her technical support in expansion culture of human cardiac fibroblasts.

## Author Contributions

**Conceptualization:** Makoto Matsuyama.

**Data curation:** Makoto Matsuyama.

**Formal analysis:** Makoto Matsuyama.

**Funding acquisition:** Takahiro Iwamiya.

**Investigation:** Makoto Matsuyama.

**Methodology:** Makoto Matsuyama.

**Project administration:** Makoto Matsuyama.

**Resources:** Makoto Matsuyama.

**Software:** Makoto Matsuyama.

**Supervision:** Makoto Matsuyama.

**Validation:** Makoto Matsuyama.

**Visualization:** Makoto Matsuyama.

**Writing – original draft:** Makoto Matsuyama.

**Writing – review & editing:** Makoto Matsuyama.

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
