## [Decision Letter · Decision Letter 0]

1 Jul 2024

PONE-D-24-22176Novel and Efficient Plasmid Transfection Protocols for Functional Analysis of Genetic Elements in Human Cardiac FibroblastsPLOS ONE

Dear Dr. Matsuyama,

Thank you for submitting your manuscript to PLOS ONE. After careful consideration, we feel that it has merit but does not fully meet PLOS ONE’s publication criteria as it currently stands. Therefore, we invite you to submit a revised version of the manuscript that addresses the points raised during the review process.

Your manuscript has been reviewed by two experts and they have expressed interest in your manuscript. They have recommended several points of improvement, all of which should be addressed prior to publication.

We look forward to receiving your revised manuscript.

Kind regards,

Keisuke Hitachi

Academic Editor

PLOS ONE

Journal Requirements:

   "I have read the journal's policy and the authors of this manuscript have the following competing interests: Takahiro Iwamiya is a cofounder and co-CEO of Metcela, Inc., and he has ownership of stocks. The corresponding author is a employee of Metcela. Takahiro Iwamiya has the authority to make payment decisions regarding employee salaries. Metcela is a company that develops VCAM1-expressing cardiac fibroblasts (VCFs) as a therapy for ischemic heart diseases. Additionally, Metcela applied for the following patents: Inventor：Takahiro Iwamiya. Assignee: Metcela Inc. Title: Composition For Injection Which Can Be Used For Treatment Of Heart Diseases And Contains Fibroblasts, And Method For Producing Fibroblast For Therapy Use. International application number: PCT/JP2018/006795. An object of the present invention is to present VCFs as a method that has not been extablished yet and that is useful for achieving long-term and fundamental cure of a necrotic cardiac tissue region to allow recovery of a heart function."

We note that one or more of the authors are employed by a commercial company: Metcela, Inc. 

Reviewers' comments:

Reviewer's Responses to Questions

**Comments to the Author**

1. Is the manuscript technically sound, and do the data support the conclusions?

Reviewer #1: Yes

Reviewer #2: Yes

2. Has the statistical analysis been performed appropriately and rigorously? 

Reviewer #1: Yes

Reviewer #2: Yes

3. Have the authors made all data underlying the findings in their manuscript fully available?

Reviewer #1: Yes

Reviewer #2: Yes

4. Is the manuscript presented in an intelligible fashion and written in standard English?

Reviewer #1: Yes

Reviewer #2: Yes

5. Review Comments to the Author

Reviewer #1: The manuscript is nice, I have some points that could help improve it:

- The Objective is not very clear. Actually, is clearer in the abstract then in the text. Try to make it more clear in the last few paragraphs of introduction.

- Instead of in the text, make a table with the primers listed in the “methods - RT-qPCR analysis”, and indicate the ones used as housekeeping.

- Its strange the Figure legends are embedded in the Results text.

- If I understood correctly, even after all improvements that actually considerably reduced toxicity and cell death, the actual transfection efficiency was less than 5% for R7 and R12 (indicated by the expression of fluorescent proteins). Please try to make this clearer in the conclusion and in the abstract. The reader should know what to expect in terms of efficiency when applying the improved fine-tailored protocol developed in this study.

- In addition, if the point above is correct (about <5% efficiency), the tittle and abstract actually does not really reflects the results, the tittle says “Novel and Efficient Plasmid Transfection….” I would say the word “Efficient” seems a bit overestimated.

Reviewer #2: The manuscript “Novel and Efficient Plasmid Transfection Protocols for Functional Analysis of Genetic Elements in Human Cardiac Fibroblasts“ provides a valuable contribution to the field of cardiovascular research by presenting a novel and efficient protocol for plasmid transfection in human cardiac fibroblasts. The authors made comprehensive evaluation of transfection conditions and the focus on reducing cell toxicity are notable strengths. However, the manuscript would benefit from addressing the concerns below, enhancing the findings' overall impact.

-For Fig 1B-D, the presentation of data on the percentage of propidium iodide (PI) positive cells is confusing. Unlike Fig1C, the % in Fig1D seems normalized to one of the controls, which demonstrated barely detectable PI+ signals and then made other groups with extremely high (>50~400 fold) changes. Using absolute percentages or normalized data based on the highest group for Fig1C would be reasonable and sufficient to solve this issue.

-‘This suggests that…’, is this a missing statement in the results section of ‘Effect of Plasmid Purification Methods on Fibroblast Viability’.

-Statistical comparisons between groups are missing in Fig 3.

-While the methods are described in detail, including specific volumes, incubation times etc., would be helpful to provide a supplementary table summarizing the optimal conditions for each reagent tested for quick reference.

-As one of the key factors/parameters that affect the efficiency of primary cell transfection, cell confluency is not explored or discussed in the study at all. Different commercial transfection reagents may have varied requirements for optimal gene expression. Meanwhile, the doubling time of HCF cells may also be considered when the actual percentage of transfected cells and cell death are calculated at the examination time point. Did the authors use the same setting for HCF confluency for all transfections and LV transduction? What’s the optimal cell confluency that can further improve the ‘best’ transfection condition identified in the study?

- The manuscript focuses on HCFs derived from a specific source/donor. It would be valuable to test the optimized protocol on HCFs from different donors or sources to ensure broad applicability. Additionally, detailed information, such as catalog number of the human cardiac fibroblast is also missing.

6. PLOS authors have the option to publish the peer review history of their article (what does this mean?). If published, this will include your full peer review and any attached files.

Reviewer #1: No

Reviewer #2: **Yes: **Yi Lin

---

## [Author Response · Author response to Decision Letter 0]

8 Aug 2024

Response to Reviewers Manuscript ID: PONE-D-24-22176 

Title: Novel and Effective Plasmid Transfection Protocols for Functional Analysis of Genetic Elements in Human Cardiac Fibroblasts

Dear Academic Editor and Reviewers,

We are grateful for the opportunity to submit a revised version of our manuscript titled "Novel and Effective Plasmid Transfection Protocols for Functional Analysis of Genetic Elements in Human Cardiac Fibroblasts." We have carefully considered the comments and suggestions provided by the reviewers and the editor, and have made the necessary revisions and additional experiments to address these concerns. Below, we provide a detailed point-by-point response to each comment.

Editor Comments:

1. Manuscript Style Requirements

We have ensured that our manuscript now adheres to PLOS ONE's style requirements. Specifically, we revised the reference format to conform to the journal's guidelines (marked-up manuscript, page 22, line 3 to page 26, line 2).

2. Funding Statement

We amended the Funding Statement to clearly declare the commercial affiliation of the founder, Takahiro Iwamiya, and specified the role of funder in our study. The updated statement includes the following:

"The funder provided support in the form of salary for author [MM], but did not have any additional role in the study design, data collection and analysis, decision to publish, or preparation of the manuscript. The specific roles of these authors are articulated in the ‘author contributions’ section."

3. Competing Interests Statement

We have updated the Competing Interests Statement to include the commercial affiliation and affirm our adherence to PLOS ONE's policies on data and material sharing. The updated statement reads:

"I have read the journal's policy and the authors of this manuscript have the following competing interests: Takahiro Iwamiya is a cofounder and co-CEO of Metcela, Inc., and he has ownership of stocks. The corresponding author is an employee of Metcela. Takahiro Iwamiya has the authority to make payment decisions regarding employee salaries. Metcela is a company that develops VCAM1-expressing cardiac fibroblasts (VCFs) as a therapy for ischemic heart diseases. Additionally, Metcela applied for the following patents: Inventor: Takahiro Iwamiya. Assignee: Metcela Inc. Title: Composition For Injection Which Can Be Used For Treatment Of Heart Diseases And Contains Fibroblasts, And Method For Producing Fibroblast For Therapy Use. International application number: PCT/JP2018/006795. An object of the present invention is to present VCFs as a method that has not been established yet and that is useful for achieving long-term and fundamental cure of a necrotic cardiac tissue region to allow recovery of heart function.

The funder provided support in the form of salary for author [MM], but did not have any additional role in the study design, data collection and analysis, decision to publish, or preparation of the manuscript. The specific roles of these authors are articulated in the ‘author contributions’ section.

The authors declare a commercial affiliation with Metcela, Inc. This does not alter our adherence to PLOS ONE policies on sharing data and materials."

4. Data Availability

We have provided all raw data for the graphs in the Figures as Supporting Information files (S1 Table and S3 File). Therefore, the statement "All relevant data are within the manuscript and its Supporting Information files." remains accurate.

5. Data Sharing Plan

We have updated the "Additional data availability information" section, unchecking all previously indicated items. We agree to make all data freely accessible in accordance with PLOS ONE’s open data policy upon acceptance of the manuscript.

6. "Data not shown" Phrase

We have removed the phrase "data not shown" and included the relevant data as S5 Fig. The corresponding changes can be found in the marked-up manuscript (page 19, line 8) and S4 Fig.

7. Original Uncropped and Unadjusted Images

We have already provided the original gel images as S4 Fig, complying with PLOS ONE's policy.

8. Complete and Correct Reference List

As mentioned earlier, we revised the reference format to meet PLOS ONE's style requirements, but the content of the reference list remains unchanged (marked-up manuscript, page 22, line 3 to page 26, line 2).

Reviewer 1 Comments:

1. Objective Clarity

We clarified the objective in the Introduction by adding a few sentences (marked-up manuscript, page 3, lines 19-29).

2. RT-qPCR Primers Table

We listed the RT-qPCR primers in a table format and clearly indicated the housekeeping genes used (marked-up manuscript, page 11, line 10).

3. Figure Legends

We followed the PLOS ONE style template, which specifies that figure captions should appear directly after the paragraph in which they are first cited.

4. Transfection Efficiency

We explicitly mentioned in the Discussion and Abstract that the efficiency of transfection using reagents R7 and R12 is less than 5% (marked-up manuscript, page 2, lines 12-15, and page 21, lines 1-3).

5. Title and Abstract Revision

We replaced "Efficient" with "Effective" in the title to better reflect our findings: "Novel and Effective Plasmid Transfection Protocols for Functional Analysis of Genetic Elements in Human Cardiac Fibroblasts." Additionally, we included the phrase "...and sufficient efficiency for functional genomic studies" at the end of the abstract to accurately convey the transfection efficiency achieved (marked-up manuscript, page 1, line 3, and page 2, lines 22-23).

Reviewer 2 Comments:

1. Fig 1B-D Data Presentation

We revised Fig 1C to use normalized data based on the highest group, as suggested by the reviewer. The revised figure is provided as Fig 1.

2. Missing Statement in Results Section

We deleted the incomplete statement "This suggests that..." from the results section (marked-up manuscript, page 14, line 22).

3. Statistical Comparisons in Fig 3

We included statistical comparison data in Fig 3. The revised figure is provided as Fig 3, and revised figure legend in the marked-up manuscript, page 15, lines 8-10.

4. Supplementary Table of Optimal Conditions

We added a supplementary table summarizing the optimal conditions for each reagent as S2 Table.

5. Cell Confluency Discussion

We provided a detailed discussion on cell confluency and its impact on transfection efficiency. This includes the optimal seeding density and its rationale. We also added a relevant section to the Discussion and included related data in S5 Fig (marked-up manuscript, page 19, lines 9-36, page 20, line 1, and S5 Fig).

6. HCF Source and Catalog Information

We added the catalog number and lot information for the HCF cells used in our optimization experiments (NHCF-V, LONZA, CAT #CC-2904, LOT #18TL281202) to the "Materials and Methods" section. Additionally, we performed supplementary experiments using HCFs from different donors and sources (KAC), and included these results in S3 Fig, panel D (marked-up manuscript, page 4, lines 3-15, and S3 Fig).

We hope that these revisions address all concerns and enhance the clarity and quality of our manuscript. Thank you for your consideration, and we look forward to your favorable response.

Sincerely,

MATSUYAMA Makoto, M.D., Ph.D.

Corresponding author

---

## [Decision Letter · Decision Letter 1]

15 Aug 2024

Novel and Effiective Plasmid Transfection Protocols for Functional Analysis of Genetic Elements in Human Cardiac Fibroblasts

PONE-D-24-22176R1

Dear Dr. Matsuyama,

We’re pleased to inform you that your manuscript has been judged scientifically suitable for publication and will be formally accepted for publication once it meets all outstanding technical requirements.

Kind regards,

Keisuke Hitachi

Academic Editor

PLOS ONE

Additional Editor Comments (optional):

Reviewers' comments:

Reviewer's Responses to Questions

**Comments to the Author**

1. If the authors have adequately addressed your comments raised in a previous round of review and you feel that this manuscript is now acceptable for publication, you may indicate that here to bypass the “Comments to the Author” section, enter your conflict of interest statement in the “Confidential to Editor” section, and submit your "Accept" recommendation.

Reviewer #1: All comments have been addressed

Reviewer #2: All comments have been addressed

2. Is the manuscript technically sound, and do the data support the conclusions?

Reviewer #1: Yes

Reviewer #2: Yes

3. Has the statistical analysis been performed appropriately and rigorously? 

Reviewer #1: Yes

Reviewer #2: Yes

4. Have the authors made all data underlying the findings in their manuscript fully available?

Reviewer #1: Yes

Reviewer #2: Yes

5. Is the manuscript presented in an intelligible fashion and written in standard English?

Reviewer #1: Yes

Reviewer #2: Yes

6. Review Comments to the Author

Reviewer #1: Now the title and abstract seems to better reflect the findings. I can now support publication of the manuscript. Also, all my points were adequately addressed in this revised version of the manuscript.

Reviewer #2: The authors have addressed my concerns raised in the initial review with substantial revisions that significantly improve the manuscript's clarity, rigor, and utility. The enhancements made, such as in data presentation, statistical analysis, and discussion of key experimental parameters, make this work a valuable contribution to the field. I belief that it will serve as an important resource for researchers working with human cardiac fibroblasts.

7. PLOS authors have the option to publish the peer review history of their article (what does this mean?). If published, this will include your full peer review and any attached files.

Reviewer #1: **Yes: **Gerson D. Keppeke

Reviewer #2: No

---

## [Editor Report · Acceptance letter]

25 Sep 2024

PONE-D-24-22176R1 

PLOS ONE

Dear Dr. Matsuyama, 

I'm pleased to inform you that your manuscript has been deemed suitable for publication in PLOS ONE. Congratulations! Your manuscript is now being handed over to our production team.

Kind regards, 

on behalf of

Dr. Keisuke Hitachi 

Academic Editor

PLOS ONE